# Learning to Dynamically Select Between Reward Shaping Signals

## Abstract

Reinforcement learning (RL) algorithms often have the limitation of sample complexity. Previous research has shown that the reliance on large amounts of experience can be mitigated through the presence of additional feedback. Automatic reward shaping is one approach to solving this problem, using automatic identification and modulation of shaping reward signals that are more informative about how agents should behave in any given scenario to learn and adapt faster. However, automatic reward shaping is still very challenging. To better study it, we break it down into two separate sub-problems: learning shaping reward signals in an application and learning how the signals can be adaptively used to provide a single reward feedback in the RL learning process. This paper focuses on the latter sub-problem. Unlike existing research, which tries to learn one shaping reward function from shaping signals, the proposed method learns to dynamically select the right reward signal to apply at each state, which is considerably more flexible. We further show that using an online strategy that seeks to match the learned shaping feedback with optimal value differences can lead to effective reward shaping and accelerated learning. The proposed ideas are verified through experiments in a variety of environments using different shaping reward paradigms.

## 1 Introduction

Although numerous successes have been reported in reinforcement learning (RL), it still suffers from several drawbacks that prevent it from performing to expectation in many real-life situations. One critical limitation is the sample complexity. In order to arrive at an acceptable solution, RL requires an enormous amount of experience (i.e., data) before useful behaviors are learned. Reward shaping is one approach that seeks to address this problem, providing additional feedback in the form of shaping rewards to allow an RL agent to learn faster. Moreover, shaping rewards that follow a potential form preserve guarantees that optimal solutions will be found despite the altered feedback (Ng et al., 1999). However, until recently, most reward shaping signals and functions have been hand-engineered. This task is notoriously difficult, as even slight incorrectness can lead to local optima that do not solve the present problem (Randløv & Alstrøm, 1998). Automatic reward shaping eliminates the difficulty of shaping reward signal and function design by learning the shaping reward function that in turn enables optimal learning of the policy.

Automatic reward shaping in itself is an extremely difficult problem to solve. In order to simplify the problem, we break down the idea into two sub-problems: (1) learning shaping reward signals, and (2) learning how to exploit shaping reward signals to provide an appropriate shaping reward at each state of the learning process. This paper focuses on the latter task, i.e., (2), which we refer to as "automatic reward adaptation".

**Problem Definition:** Given a set of shaping reward signals $\vec{\phi} = \phi_1, \ldots, \phi_n$, learn to adapt these signals automatically to produce a single shaping reward $F(s, a, s') : \vec{\phi} \to \mathbb{R}$ for each state $s$, action $a$, next state $s'$ tuple in the learning process. The full reward for any transition in the RL problem is $R(s, a, s') = r(s, a, s') + F(s, a, s')$, where $r$ is the original reward of the RL problem before shaping.

Our proposed approach to automatic reward adaptation is to learn to dynamically select the right shaping reward signal $\phi_i$ from $\vec{\phi}$ at each transition encountered in learning. In addition, our method

learns using minimal infrastructure and with value-based gradients. We avoid the use of models and additional approximate value functions (which previous approaches to automatic reward shaping typically rely on) and perform updates using already-present values as feedback and with minimal additional computation. The proposed ideas have been verified through experiments in a variety of environments using different shaping reward paradigms.

The basis of our shaping signal selection approach is rooted in how humans seem to react in realistic decision-making situations. Given a set of basic reward signals, say "comfort" and "self-preservation", humans have the uncanny ability to identify when and how much one should listen to any given signal depending on the current situation. For example, in a lane keeping task, if we were near the edge of a lane with no cars around us, we would simply listen to the "comfort" reward signal telling us to move closer to the center of the lane. However, if there was another car moving into the same lane at the same time, we would instead follow the "self-preservation" reward signal dictating that we stay as reasonably far away from other cars as possible. Both signals lead to correct performance but are applicable in entirely different situations, provide different information, and, most importantly, induce different behaviors. While we recognize that explicit selection risks not using available information in other shaping reward signals, we argue that it provides a guaranteed improvement over only the environment reward. In this sense, selection does not hinder the RL agent in learning even though it is incomplete relative to optimal reward shaping, which is extremely difficult to design.

This work parallels the area of multi-objective RL (MORL), which investigates how to perform RL in the presence of multiple rewards (Roijers et al., 2013). However, there are a couple differences between our idea and previous work in automatic reward shaping and MORL. First, much research in automatic reward shaping focuses on the first sub-problem (1) described above, i.e., learning the parameters of some shaping reward signal (usually only one). However, realistic problems often involve a number of (possibly conflicting) goals or signals that cannot be trivially summarized as a single signal or function. Our work focuses on the second sub-problem (2) and investigates if there are better ways to access provided shaping reward signals for more effective reward shaping. Second, while MORL also handles environments with multiple feedback signals, it aims to solve a different problem overall. MORL attempts to learn solutions that best optimize the multiple rewards presented to it, but our idea attempts to best exploit multiple rewards in a way that solves the single objective present in the problem. Furthermore, MORL typically learns a linear combination of signals as the final reward shaping function. While effective in many cases, a learned fixed combination does not always suit each individual state in learning, especially if the environment is dynamic. That being said, we do not discount the potential of combination in automatic reward adaptation.[1] Our goal is simply to consider another approach with promising flexibility.

## 2 RELATED WORK

The fundamental theory behind reward shaping regarding its optimality-preserving guarantees was documented in Ng et al. (1999). Since then, the guarantees of potential-based reward shaping have been expanded to time-varying potential options by Harutyunyan et al. (2015), which more naturally reflects realistic problems and the concept of automatic reward shaping. One of the initial works in this area demonstrated impressive results but the shaping reward function was heavily engineered (Laud & DeJong, 2002). More recent research has made progress towards greater autonomy. The mechanisms driving automatic reward shaping typically fall into a few categories, with one of the originals involving the use of abstractions to learn values on a simpler version of the problem before using these values as shaping rewards (Marthi, 2007). Other projects have followed up this idea using both model-free and model-based RL methods (Grześ & Kudenko, 2010) as well as with the estimated modeling of the reward function itself (Marom & Rosman, 2018). One particular work directly bootstraps the model-based learning in R-max into a shaping reward function (Asmuth et al., 2008). Yet another builds a graph of the environment before using subgoals as a means of defining shaping rewards which adjust as the graph is updated (Marashi et al., 2012). Credit assignment is another form of automatic reward shaping. It injects information into previous experiences such that learning is enhanced when replay occurs. Song & Jin (2011) implemented this by identifying critical states and using these landmarks as sub-rewards to make learning easier. De Villiers &

---

[1]We will study flexible combination of individual shaping signals in our future work.

Sabatta (2020) directly augmented the replay buffer with propagated rewards to make the reward surface more dense and accelerate learning. Zheng et al. (2018) integrated learning of an intrinsic motivation signal along with the agent's value and policy learning. While representative of automatic reward shaping, they only consider the single intrinsic motivation feedback signal when performing reward shaping.

Adjacent to our focus is the area of multi-objective RL (MORL). Brys et al. (2014) explored some connections between multi-objective frameworks and reward shaping which Brys et al. (2017) later expanded on by directly characterizing shaping reward signals as multiple objectives and applying MORL to successfully solve such problems. Van Seijen et al. (2017) implemented a form of automatic reward adaptation by learning a separate value function for each reward signal and using a value estimated from each reward signal's associated value to update the policy. Fu et al. (2019) addressed the presence of multiple goals (in the form of rewards) by explicitly calculating the best weighting for each reward in a linear combination after some amount of experience. Tajmajer (2018) similarly used a linear combination to combine multiple rewards but simultaneously learned a set of decision values that acted as weights for each reward in the final feedback computation. As mentioned previously, our work differs from these methods primarily in that our method does not use linear combination, uses less learning infrastructure, and aims to control reward shaping (not the agent itself in the pursuit of multiple goals). Instead of learning the weights for rewards in solving a problem with multiple objectives, we dynamically select between shaping reward signals to use in learning at any given point. While inspired by concepts in MORL, this selection technique is, to our knowledge, new.

## 3 BACKGROUND

### 3.1 REINFORCEMENT LEARNING

RL is a form of machine learning that specializes in learning to solve problems that involve unknown environments and require sequences of decisions. Such problems are often characterized as Markov Decision Processes (MDPs), and described by a tuple M = (S, A, $\tau$, R, $\gamma$). S represents the set of states, A is the set of possible actions, $\tau$ is the transition probability function $\tau : S \, x \, A \rightarrow S$, and R is the reward function $R : S \, x \, A \, x \, S \rightarrow \mathbb{R}$. $\gamma$ is a parameter that describes the discount factor. MDPs operate generally through the following repeating set of steps: a state is observed, an action is taken which transitions the agent into another state while a reward is given by the environment, then the transition and reward are used to update the policy and the value function (if used).

The goal of any given agent attempting to solve an MDP is to generate a policy through interactions with the environment that maps states to actions, $\pi : S \rightarrow A$. This policy is optimized towards the objective of maximizing the expected sum of discounted rewards $G = \mathbb{E}_\pi[\sum \gamma r \mid S = s, \, A = a]$, or return. Q-learning is a well-established algorithm that follows the Q-value, or action-value, form of this objective by attempting to find the action-value function $Q(s, a) = \mathbb{E}_\pi[G \mid s, \, a]$ (Kröse, 1995). Updates take the following form:

$$Q(s, a) = Q(s, a) + \alpha(r + \gamma \max_a Q(s', a) - Q(s, a)) \qquad (1)$$

with $\alpha$ as the learning rate (Sutton & Barto, 2018). Typically, some form of encouraged exploration is implemented, most often by specifying a probability (often decaying over time) with which the agent must select a random action instead of the greedy one.

### 3.2 REWARD SHAPING

In the above update equation (Eq. 1), the reward r specified is the only feedback provided by the environment. Reward shaping seeks to amplify this feedback by providing more information, often referred to as the shaping reward F. The inclusion of this term intuitively leads to the augmented update equation:

$$Q(s, a) = Q(s, a) + \alpha(r + F(s, a, s') + \gamma \max_a Q(s', a) - Q(s, a)) \qquad (2)$$

Note that F is reliant on the current state, the taken action, and the next state. The shaping reward can be formulated such that it has guarantees of convergence to the original MDP's optimal policy

if the shaping reward function follows the potential shaping paradigm. A shaping reward F is in potential-based form if it takes the difference of any shaping reward signal $\phi$, accounting for the discount factor:

$$F(s, a, s') = \gamma\phi(s') - \phi(s) \tag{3}$$

Of note is the fact that the optimal shaping reward function is exactly the optimal value function (Ng et al., 1999).

## 4 METHODS

### 4.1 AUTOMATIC REWARD ADAPTATION

Before we describe our method of representing reward signals, we argue for an intuitive and general shaping reward format. Assume that an agent encounters a set of additional shaping reward signals $\vec{\phi}$. These signals $\vec{\phi}$ should imitate human instincts (e.g. "comfort" or "self-preservation") and thus be able to be easily formulated for many problems. This assumption is justified because an agent attempting to solve an RL problem must be provided with a manually designed feedback at some level, whether that be in the form of the environment reward function or through a set of shaping reward signals. Additionally, this assumption enables improvement because it allows for the specification of intuitive rewards that domain experts can more easily design.

According to Ng et al. (1999), the optimal form of any shaping reward signal is exactly the optimal value function. Thus, learning an optimal shaping reward function is equivalent to finding the parameters $\eta$ that give shaping rewards closest to the optimal values. The goal then is for all shaping reward signals $\vec{\phi}$ to attempt to approximate at least a component of the optimal value function, and for the potential-based difference in these shaping reward signals $\gamma\phi(\vec{s'}) - \phi(\vec{s})$ to attempt to approximate the shape of the optimal value surface (i.e. the difference in optimal values). However, optimal values, and of course the optimal value function, are unknown in any RL problem (and indeed are the motivation for RL). Options for estimating optimal values include receiving explicit knowledge from a (typically external) expert or through experience. Assuming the presence of an expert in realistic applications is unreliable, and thus the most robust option is to attempt to learn the shaping reward function through experience. The following selection method, parameterized by $\eta$, is updated according to these tenets. Adjustments are made towards minimizing the difference between potential-based shaping reward signals and the difference in approximated values $(r + \gamma \max_a Q(s', a) - Q(s, a)) - (\gamma\phi_i(s') - \phi_i(s))$. The convergence guarantees of RL given by general policy iteration (Sutton, 2000) hold in this augmented learning procedure given that the shaping reward signals do not override the optimal values. This has been previously shown with respect to reward shaping by Brys et al. (2017) and can be achieved by scaling of the shaping reward signal differences.

### 4.2 SELECTION

Selection in the context of automatic reward adaptation describes a procedure in which some mechanism, henceforth referred to as the "selector", makes an exclusive decision regarding which shaping reward signal to listen to. A selector can be represented as a function approximator that receives as input the current state $s$ and the taken action $a$. The output is a softmax set of probabilities that describes which portion of the current shaping reward signals to use in characterizing the shaping reward value $F$.

$$F(s, a, s') = \mathbb{1}_{Softmax(\vec{c})_i=1} \cdot (\gamma\vec{\phi}(s') - \vec{\phi}(s)) \tag{4}$$

Shaping reward signals are again used in the potential-based form. The processing of these signals through the selector does not invalidate the optimal policy guarantees, as seen in this equivalence:

$$\begin{aligned}
F(s, a, s') &= \mathbb{1}_{Softmax(\vec{c})_i=1} \cdot \vec{c} \cdot (\gamma\vec{\phi}(s') - \vec{\phi}(s)) \\
&= \gamma(\mathbb{1}_{Softmax(\vec{c})_i=1} \cdot \vec{c} \cdot \vec{\phi}(s')) - \\
&\quad (\mathbb{1}_{Softmax(\vec{c})_i=1} \cdot \vec{c} \cdot \vec{\phi}(s)) \\
&= \gamma\vec{\phi'}(s') - \vec{\phi'}(s)
\end{aligned} \tag{5}$$

The selector is updated according to the previously-described principle of differences in shaping reward signals approximating the differences in the optimal value function across corresponding transitions. Each signal's distance from the estimated value difference is calculated using $(r + \gamma \max_a Q(s', a) - Q(s, a)) - (\gamma \phi_i(s') - \phi_i(s))$. In addition to the value distance, an augmenting factor can be included to better guide the selector. Note that the magnitude of a difference in shaping reward signals $|\phi_i(s') - \phi_i(s)|$ represents some measure of importance in the associated shaping reward signal. This factor can be used directly in the update to adjust the selector towards signals that potentially carry more significant information. Overall, the augmented distance takes the following form:

$$\frac{|(r + \gamma \max_a Q(s', a) - Q(s, a)) - (\gamma \phi_i(s') - \phi_i(s))|}{(|\phi_i(s') - \phi_i(s)| + c)} \qquad (6)$$

The $c$ term is a small constant to avoid divide-by-zero errors. The shaping reward signal with the smallest distance is considered to be the best decision and set as the target for the selector update. Categorical cross-entropy is then applied as the loss function between the selector output probabilities and the target vector. Algorithm 1 outlines a basic implementation of selection in automatic reward adaptation. Note that this implementation of selection is inherently dynamic as it

---

**Algorithm 1:** Automatic Reward Adaptation RL - Selection

---

1   Initialize: policy $\pi$, Q-value function $Q$, and selector $\Sigma$
2   **while** *not converged* **do**
3      Sample action $a$ from policy $\pi$
4      Take action $a$, observe reward $r$ and new state $s'$
5      **for** *experience* $(s, a, r, s')$ *in replay buffer* **do**
6          Sample selector softmax probabilities $c$ from $\Sigma$
7          $F(s, a, s') = \mathbb{1}_{Softmax(\vec{c})_i = 1} \cdot \vec{c} \cdot (\gamma \vec{\phi}(s') - \vec{\phi}(s))$
8          $Q(s, a) \leftarrow Q(s, a) + \alpha(r + F(s, a, s') + \gamma \max_a Q(s', a) - Q(s, a))$
9          $diffs_i = \frac{|(r + \gamma \max_a Q(s', a) - Q(s, a)) - (\gamma \phi_i(s') - \phi_i(s))|}{(|\phi_i(s') - \phi_i(s)| + c)}$
10         $\vec{c} = \mathbb{1}_{argmin(di\vec{f}fs) = i}$
11         $\Sigma \leftarrow \vec{c}$
12      **end**
13      $s = s'$
14   **end**

---

determines which shaping reward signal to prioritize based on the current state and action. As an augmentative component within the agent's learning infrastructure, the selector constantly adjusts to experienced transitions and provides adaptable shaping feedback online.

## 5   EXPERIMENTS

The selection paradigm was tested on a variety of baselines provided by MuJoCo (Todorov et al., 2012) and OpenAI Gym (Brockman et al., 2016). Reacher was used from MuJoCo and required the agent to maintain the tip of a robotic arm close to some goal point in a 2D plane. From Gym, CartPole, Pendulum, and MountainCar were selected for their diversity in reward functions. In CartPole and Pendulum, the agent must balance a linkage in the vertical up position. MountainCar is a challenging task that requires an agent to control a car with the goal of reaching the top of a slope.

Shaping reward signals were defined for each of the environments in such a way that a variety of different shaping potential surfaces could be tested. For CartPole, each signal rewarded the agent for keeping the corresponding state component, either cart position or pole angle, near zero. For Pendulum, the first shaping reward signal rewarded the agent purely for how close the pole was to vertical up. The second shaping reward signal accounted for both the pole angle and velocity and was the more accurate signal. The MountainCar environment shaping reward signals tested similarly for distinction capabilities in the selector. The first shaping reward signal rewarded horizontal position of the car and was suboptimal. The second shaping reward signal rewarded for a combination of

Table 1: Experimental environments and shaping reward functions

| Environment | Shaping Functions |
|---|---|
| Gym - CartPole | $\phi_1 = -10 \cdot pole\_angle$ 
 $\phi_2 = -cart\_position$ |
| Gym - Pendulum | $\phi_1 = \cos(pole\_angle)$ 
 $\phi_2 = -\|pole\_velocity\| \cdot \cos(pole\_angle)$ |
| Gym - MountainCar | $\phi_1 = car\_position$ 
 $\phi_2 = \|car\_velocity\| \cdot 10 + car\_position/10$ |
| MuJoCo - Reacher | $\phi_1 = \begin{cases} -distance & distance \leq 1.5 \\ 0 & distance > 1.5 \end{cases}$ 
 $\phi_2 = \begin{cases} 0 & distance \leq 1.5 \\ -distance & distance > 1.5 \end{cases}$ |

car horizontal position and velocity and was effective at leading the agent to the solution. For the Reacher environment, each shaping reward signal provided information leading to the goal point but in different regions of the state space. The selector was required to learn when to listen to each signal. Further mathematical details are described in Table 1.

For Reacher, Pendulum, and MountainCar, DDPG with decaying noisy actions for exploration was used for its continuous control capabilities and competitiveness in many classic RL problems (Lillicrap et al., 2016). A version of DQN with no target network and epsilon-greedy exploration was used for CartPole (Mnih et al., 2015). Neural networks were used for all function approximation implementations. For the sake of space, hyperparameters for each environment are documented in the Appendix. To account for randomness, seeds were used to control random parameters in the neural networks as well as in the exploration noise.

Performance was measured as the number of episodes until the averaged total reward over the past 50 episodes was maximized. Implementations of un-augmented algorithms were included as a basic comparison to standard learning (denoted as the "base" agents). In addition, a competitive comparison was included in the form of an agent that learned from a direct linear combination of provided shaping reward signals (denoted as the "shaped" agents). Experiments were run five times, with the experiment showing the best performance taken and presented in the figures.

Comparisons were made to the recent automatic reward adaptation method described in Fu et al. (2019), which explicitly calculated best weightings of shaping reward signals. Their tests were performed in two similar environments, CartPole and MountainCar, and we applied our selection technique to their shaping reward signals in each problem.

All experiments were run on one of two setups. The first was a Windows 10 Home laptop containing 16 GB of memory, an Intel i7-9750H CPU, and an NVIDIA GeForce GTX 1650 graphics card. The other laptop ran Ubuntu 16.04 and contained 8 GB of memory, an Intel i7-4819MQ CPU, and an NVIDIA GeForce 840M graphics card. All experiments were run on the GPUs when possible. Experiment programs were written in Python 3.6 and heavily utilized the following libraries: TensorFlow 2.1.0, gym 0.15.4, mujoco-py 1.50.1.0. Code implementing the ideas detailed in this work can be found at https://github.com/.

## 6 DISCUSSION

Visualization of the convergence speeds for all tests are shown in Figure 1. In all environments, the selection agent outperformed the base agent and typically matched the shaped agent's performance closely. In each environment, the selection agent was able to identify which shaping reward signal(s) to listen to and use those signals to reinforce correct behaviors. This can be particularly clearly seen in the Reacher and MountainCar environments. In Reacher, the selection agent correctly identified the shaping reward signals in each region leading the agent to the goal point. In MountainCar, the selector correctly identified the optimal shaping reward signal ($\phi_2$) and ignored the poor shaping reward signal ($\phi_1$). The base agent was unable to find the solutions to either of these problems and

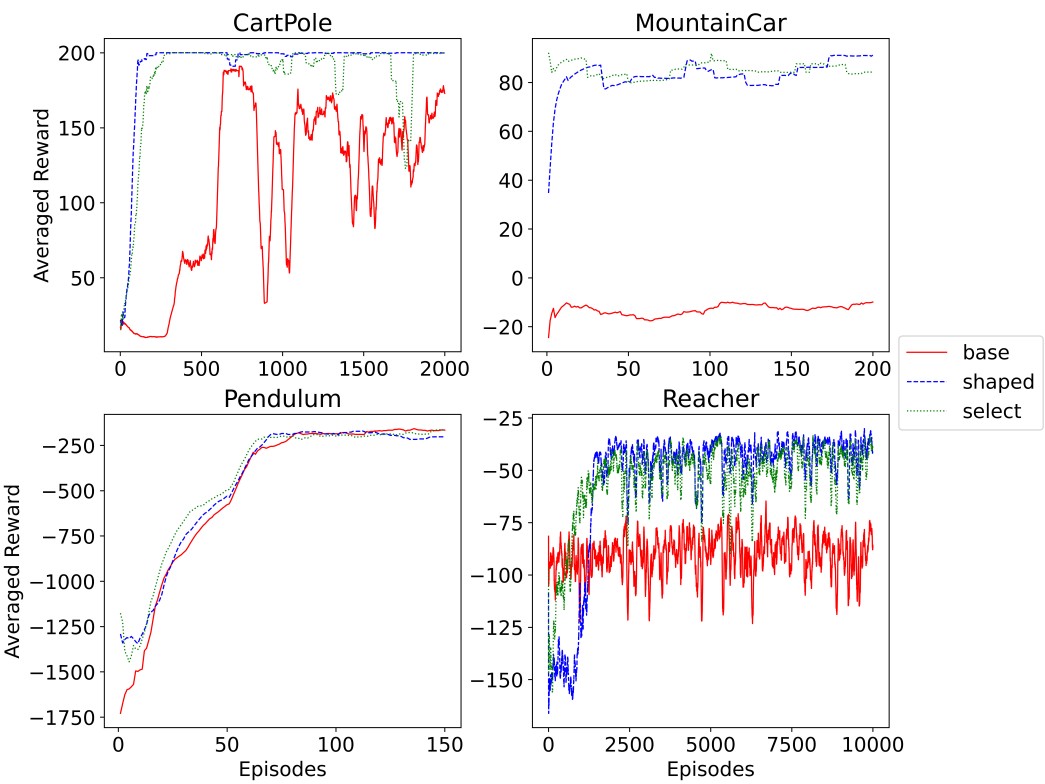

Figure 1: Performance results of the selection method against base and shaped agents

got stuck in a local optimum of not moving at all. The CartPole environment shows that selection is able to maintain optimal solutions despite perturbations in learned behaviors. The selection agent was able to quickly re-converge to the solution each time a fluctuation occurred. This also underlines the dynamic nature of our selection method for automatic reward adaptation. In response to poor behaviors encountered during learning, the selector was able to re-discover the correct shaping reward signals and re-learn the optimal solution. The Pendulum environment was quickly solved by all three agents. It acts as a baseline check for the selection method and demonstrates that selection does not lead the agent to other solutions with suboptimal performance.

The Fu versions of CartPole and MountainCar used slightly different shaping reward signals. For CartPole, they used signals for cart position and pole angle that were similar to ours except for small value scaling differences. For MountainCar, they defined signals for cart position, velocity, and height. Each signal rewarded transitions leading to higher corresponding values. It should be noted that the Fu version of MountainCar used a different reward function that gave a constant small negative reward for all time steps. However, the goal and optimal policy remain the same for this version of the problem. For the comparison experiments, we applied our selection method in the Fu versions of each environment and using their defined shaping reward signals. Effectiveness of our method and Fu's method in accelerating learning are shown in Figure 2. Convergence speeds are more explicitly details in Table 2. In this case, convergence was defined as the earliest episode in which the maximum averaged reward was observed.

Table 2: Episodes until convergence for each automatic reward adaptation method (value of converged reward in parentheses)

|  | Selection | Fu |
| --- | --- | --- |
| CartPole | 602 (1000.0) | 849 (995.31) |
| MountainCar | 862 (-117.27) | 964 (-111.62) |

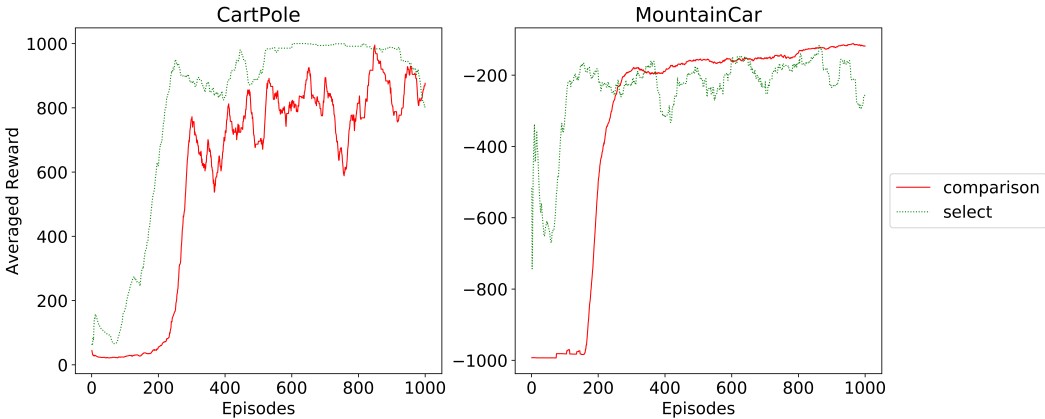

Figure 2: Comparisons in performance between our selection idea and the automatic reward adaptation method presented in Fu et al. (2019)

When applied to these versions of the problems, the selector was still able to identify the correct shaping reward signals to listen to. The selection agent converged to the correct policy in both the CartPole and MountainCar problems, often faster than the Fu agent. In CartPole, selection arrived at a better solution than Fu earlier in learning. In MountainCar, selection learned a policy that performed slightly worse than that learned by Fu's method, but discovered it earlier. Later in learning, the selection agent maintained competitive performance despite lacking the extra time to compute optimal usage of available shaping reward signals. After reaching convergence, the selection agent's performance remained close to that of the Fu agent. While there was some fluctuation in the policy, the selection method was consistently able to adjust the agent back towards the optimal policy. Overall, our version of automatic reward adaptation competed respectably even when learning with less infrastructure.

## 7 CONCLUSION

Automatic reward adaptation is a newly considered component of automatic reward shaping that focuses on accelerating reinforcement learning through the dynamic processing of multiple available shaping reward signals. Design was focused on adjusting the selector towards better approximating changes in the optimal value function. By following the learned value function, the selected shaping reward function could more quickly provide information that pushed the agent towards the optimal solution. Furthermore, through the use of these value-based updates, the selector is free of excess external interaction and minimizes offline computation. These properties potentially promote our implementation of automatic reward adaptation as a step toward RL that is more applicable to realistic problems.

There are many paths forward from this point. First and foremost is the other component of automatic reward shaping: automatic learning of reward signals themselves. While there has been work investigating how to approach this idea, much research remains to be done regarding how these signals can and should be characterized, how they should be bounded (if at all), and how we can generalize the learning procedures to different domains, just to name a few paths. In addition, there are many more ways of selection (and combination) that can be explored that have not been analyzed in this work. We intend to continue pursuing automatic reward shaping ideas in future work.

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

## A  APPENDIX

Table 3: Performance Hyperparameters for each Environment

|  | CartPole | Pendulum | MountainCar | Reacher |
|---|---|---|---|---|
| Critic/DQN Learning Rate | 0.001 | 0.002 | 0.002 | 0.002 |
| Actor Learning Rate | N/A | 0.001 | 0.001 | 0.001 |
| Selector Learning Rate | 0.002 | 0.01 | 0.003 | 0.003 |
| Target Network Update Rate | N/A | 0.005 | 0.005 | 0.005 |
| Discount Factor | 0.95 | 0.99 | 0.99 | 0.95 |
| Exploration Factor | 0.9 | 0.2 | 0.9 | 0.2 |
| Exploration Factor Decay Rate | 0.95 | N/A | N/A | N/A |
| Batch Size | 64 | 64 | 64 | 64 |
| Replay Memory Size | 50000 | 50000 | 50000 | 50000 |
| Optimizer | Adam | Adam | Adam | Adam |

DDPG used the Ornstein-Uhlenbeck process for exploration noise generation. Thus, the Exploration Factor was actually the standard deviation for the process instead of the standard epsilon factor used in, for example, DQN. For the same reason, the Exploration Factor Decay Rate for DDPG was listed as N/A.

