# OpenReview forum: "Learning to Dynamically Select Between Reward Shaping Signals"
_ICLR.cc/2021/Conference — Reject_

### Official Review · AnonReviewer3 · 2020-10-27
**weak experiments, missing CI and baselines**

**Rating:** 5
**Confidence:** 3

**Review:**

The paper presents an approach to select the best reward shaping potential signal out of multiple available shaping potentials. The main idea seems to be to select the shaping signal that minimizes the inverse of the difference of potentials between the next state and the current state. The experiments show the proposed approach works better than other baselines.

- How is the proposed approach different from selecting the potential function that locally offers the highest increase in potential?
The proposed augmented distance that decides the selector is = (TD-error - potential-difference)/potential-difference = TD-error/potential-difference. Since we are sampling from experience replay, the s' is fixed which means TD-error is fixed across all potential functions. So the only quantity that changes is potential-difference_i for i in all potential functions. Is it not the same as picking the shaping reward that is the greatest?
What if one of the potential signals is not conducive to learning the original Q. Would that not slow down learning in this case?

- I believe the main weakness of the paper is the experiment sections. The main baseline that the paper uses is a linear combination of multiple potential functions. It would have been much preferable to see the performance of various potential functions separately (not as a linear combination). That would have shown how if the proposed approach is able to select the best potential function out of the available potential functions.
Table 1 does not say how the proposed shaping functions are potential functions?
Each experiment could add one more potential function that is negative of the best potential function. This way the effect of the proposed approach could be highlighted, showing how quickly/slowly the proposed approach was able to ignore the negative of good potential function since its not really helping with the learning.
The experiments currently are missing confidence intervals / std. error bars.

- The paper is quite unclear. It is not clear what the paper means by optimal value difference, the use of phi is unclear. In the project definition, it is called shaping reward signal. It is only later that the reader realizes that phi is a potential function.
So the paper assumes that various potential functions are available as shaping advice. Potential functions are generally defined over the whole state-action pairs. How is the proposed approach applicable when the state-action space is large? What if it is not possible to define an exact potential function over the large state-action space.
The paper provides a literature review of multi-objective RL but it is not clear how it is connected to MORL. At certain point in the paper it seemed that the problem of selecting among various shaping potentials will be cast as a MORL but that does not seem to be the case.
Some minor notation mistakes: In Background: R: SxAxS . I believe the paper meant to use \times  and not $x$.
The variable could be introduced before it is used in equation 5. Only later in the pseudocode, the reader realizes what c is.
The above mistakes make it quite difficult to understand the complete picture presented by the paper.

---

### Official Review · AnonReviewer1 · 2020-10-27
**Review of Learning to Dynamically Select Between Reward Shaping Signals**

**Rating:** 2
**Confidence:** 5

**Review:**

In this paper, the authors present an exploration of learning reward signals and how various reward signals can be adaptively used to provide a single reward feedback channel for a reinforcement learning process. First, they provide a solid introduction and related work research on multi-objective reinforcement learning. I would urge the authors not to present statements about work that they will continue to work on or work towards in the future. These forward-looking statements do not serve the purpose of the current work.

A great deal of real estate in this paper is committed to background information on related work reinforcement learning and reward shaping which does not serve this paper. Indeed it is not until section 4.2 Methods: Selection when the author's finally present details on new methods and experimental design.

The results and figures presented are unclear and I would urge the authors to reconsider the information that they would present in terms of mean and measure of central tendency. It is hard to compare the significance of their results against their baseline methods.

This paper feels incomplete to me, and I would like to see this research continued toward a more full submission. Finally, I do not feel as though this paper is suitable for the general audience at ICLR. Perhaps there would be audiences more interested in the specific application and innovations in reward shaping at other workshops.

---

### Official Review · AnonReviewer2 · 2020-10-28
**Learning to dynamically select between reward shaping signals**

**Rating:** 4
**Confidence:** 4

**Review:**

Summary of review:
Intruiguing idea, but the core of it is not properly exposed, and the empirical methodology has some flaws.

Description:
This paper seeks to leverage the advantage conferred by reward shaping signals in accelerating RL learning, by adaptively selecting from give shaping signals.

The main novelty of the work is in the use of a selector to adaptively choose which reward shaping signal to use.

Strengths:
-	Reward shaping is a proven method to accelerate learning. It is worth exploring more flexible and adaptive mechanisms for reward shaping.
-	The proposed method is simple to implement and understand, and presumably to monitor during training.

Weaknesses:
-	The question of how the shaping signals are selected / designed is not addressed directly.  When I started reading the paper, I expected this to come from transfer (from similar domains), but in the experiments it seems to be hand-designed.
-	The specific mechanism to adjust the selector is obscure.  Parsing through algorithm 1, what is the index “i”?   In addition to a selector, do you assume (line 1 of Alg.1) that you also receive a finite set of shaping signals?  I have a rough idea, but it is not clear from the paper.
-	What are the properties of this selection mechanism, in terms of convergence of the algorithm?  Does it converge? And under what conditions on the selector would the solution converge to Q()?
-	The empirical results as presented do not meet the standards for good scientific reporting.  There is no standard error on any of the results.  The number of seeds is not reported.  Therefore, I do not have confidence in the results.  Furthermore it’s not clear in Fig.1 that the selected shaping signal is any better than the shaped reward.

---

### Official Review · AnonReviewer4 · 2020-10-28
**a practical idea, but some implementation details remain unclear, and experiments are not conclusive**

**Rating:** 4
**Confidence:** 4

**Review:**

Summary:

The idea of potential-based reward shaping (PBRS) is to improve the performance of learning agents by incorporating additional domain knowledge into their reward function (making it dense), while maintaining the same asymptotic convergence guarantees. However, one aspect of PBRS, that is introduced in this work, is how to select among multiple shaping signals in an adaptive manner that is also dependent on the current context, in a way that can overall improve the performance of the agent. A method is proposed  here that learns to select among two or more shaping functions based on experience, represented as the difference between the TD error (or surprise) and the shaped reward signal, the conjecture being that signals that are closer to the TD error can lead to the best improvement in the agent's performance. The signal with the smallest difference is used as a self-supervised label to train a classifier that selects the signal. Experiments show that the method is at par with other similar approaches while being simpler to implement.

Pros:
- the problem setting is one of high practical significance, since we may often be presented with various rewards but not know how to use them, and an adaptive approach for selecting between them on the basis of data seems logical
- the approach seems quite versatile and easy to implement (if my understanding of the workflow is correct), and appears to be compatible with most standard RL algorithms that use a notion of Q-values
- in principle, as the selector is running entirely online, could be applied in non-stationary RL setting (more comments below). an interesting extension could indeed be had in future work as the authors suggest.

Conceptual issues:
- my main concern with the paper is the claim that the proposed update (6) when used in the context (5) maintains policy invariance. More specifically, the softmax probability over signals, c, depends on the current action (and in fact the history of data) through (6). However, this dependency is not shown in (5). While existing work validates the action-dependence of reward shaping signals, it requires that the potential function be applied in a time consistent and look-ahead manner, and requires the use of the biased-greedy policy (this being because action dependence shifts the Q(s,a) values non-trivially by phi(s, a) and requires it to be corrected). I do not immediately see how the action dependence is handled in a time consistent manner as predicated in the previous work (Wiewiora, 2003). I do admit that in practice, these issues can often be difficult to deal with, particularly with the use of experience replay and an episodic setting. It would be highly beneficial to elaborate on how you were able to address these theoretical issues to maintain policy guarantees? Even if the c is only dependent on state, I believe it should be applied in a time consistent manner to avoid instability, by caching the c from the last iteration and applying it to each phi individually (this would be the dynamic reward shaping work of (Devlin, 2012)).
- the link between the importance of the shaping signal and the difference in Q values as defined by the TD error (6) is not well discussed. One of the fundamental drawbacks of this approach is that it could be scale sensitive with respect to phi; for instance, let's say that the correct phi_i were scaled by 1000, then why would (6) be able to select the right signal in this case? it seems the TD error would remain constant but the right term in the numerator would increase, making the error larger and forcing it towards the maximal value it could take.
- the authors suggest that the method should work when the signals provided are adversarial in nature, but the experimental setup does not include adversarial examples, as all the potentials seem useful in different contexts. for example in cartpole, phi1 could be useful when the pole is tilting, while phi2 could be useful when the cart is about to leave the frame. a missing detail is how robust the method is to adversarial signals. Since the TD error is quite noisy, how to avoid the situation in which the signal may be similar to the TD error but the signal is adversarial or not useful?
- Reinforcing the previous point, can some further intuitive (or even better, theoretical) derivation be provided from first principle to validate why (6) is the right way to select signals? This would strengthen the technical aspects of the paper that could be quite improved at this stage.
- while the authors point out that the approach of selecting signals based on the primary value function is new, a search reveals some existing work (e.g. Gimelfarb et al., 2018) that used values to select signals, which appears superficially similar to this work. How would the proposed method compare to a non-context specific approach such as this that seems to also use values when comparing signals?

Experimental details:
- the authors mention the comparison to a shaped agent. is this the same as using the learned c as a weighting over signals, rather than picking the best signal from the selector output? if so, the results for the weighting appear more stable than the selection of a single signal, so why do the authors not use this weighted combination rather than picking one as the contribution? the selection of a single signal seems less robust to learning errors than a weighting, since it may be brittle and cause instability when the single best expert (chosen by arg max) must suddenly switch.
- the experiments do not reveal the true benefit of selecting different signals in a state-dependent way, which seems to be the main contribution. for example, how do the results compare to a single signal? also, for the lower dimensional tasks (cart pole, mountain car), it would be really helpful and interesting to illustrate which signal is chosen and how this selection process is changing over time and situation. Given that the signals are chosen so that one should be clearly optimal in different situations, this will validate the claim that adapting the signal to different situations is useful and working as intended.
- the comparison to Fu et al, 201`9 is rather inconclusive and incomplete. the results on these problems do not indicate that the proposed method advances the state of the art. The authors do mention that the method is easier to implement and run. Does this mean that the expected or discount reward is not the right measure to use or does not reveal the full story? Is computation time, or memory use, a more suitable measure for comparison? If so, this could be one selling point that will strongly benefit the paper.
- the proposed method seems to have some instability in some situations. while reporting the best case is useful, it does not reveal the full story. I think the paper could be made clearer by reporting the average or median performance with confidence interval bars, and discussing how learning stability is affected by these methods.

Other details:
- the authors use c twice, one for the underflow correction and one for the softmax

Overall, I think this paper is quite interesting, well motivated and could be of significance to the broad RL community. However, I would like to see more discussion, and given this method is largely empirical, a more thorough analysis of where the method works and what metrics are most suitable to evaluate it.

References:
- Devlin, Sam Michael, and Daniel Kudenko. "Dynamic potential-based reward shaping." Proceedings of the 11th International Conference on Autonomous Agents and Multiagent Systems. IFAAMAS, 2012.
- Gimelfarb, Michael, Scott Sanner, and Chi-Guhn Lee. "Reinforcement learning with multiple experts: A bayesian model combination approach." Advances in Neural Information Processing Systems. 2018.
- Wiewiora, Eric, Garrison W. Cottrell, and Charles Elkan. "Principled methods for advising reinforcement learning agents." Proceedings of the 20th International Conference on Machine Learning (ICML-03). 2003.

---

### Decision · Program_Chairs · 2021-01-07
**Final Decision**

**Decision:**

Reject

**Comment:**

This paper presents an approach to reward shaping in RL centred on the question of how to select between different shaping signals. As such this is an interesting research direction that could make important contributions in the area.
Generally the reviewers felt that the paper is too preliminary in its current form. There were several questions raised around problems with the technical formulation. It was also felt that the experiments could be more rigourous to fully validate the claims of the paper.